# Estimating repeat spectra and genome length from low-coverage genome skims with RESPECT

**Shahab Sarmashghi**[1], **Metin Balaban**[2], **Eleonora Rachtman**[2], **Behrouz Touri**[1], **Siavash Mirarab**[1], **Vineet Bafna**[3]*

**1** Department of Electrical & Computer Engineering, University of California, San Diego, La Jolla, California, United States of America, **2** Bioinformatics & Systems Biology Graduate Program, University of California, San Diego, La Jolla, California, United States of America, **3** Department of Computer Science & Engineering, University of California, San Diego, La Jolla, California, United States of America

* vbafna@cs.ucsd.edu

**Data Availability Statement:** RESPECT software and the trained models are publicly available on https://github.com/shahab-sarmashghi/RESPECT under a BSD 3-Clause license. Run accessions of

## Abstract

The cost of sequencing the genome is dropping at a much faster rate compared to assembling and finishing the genome. The use of lightly sampled genomes (genome-skims) could be transformative for genomic ecology, and results using $k$-mers have shown the advantage of this approach in identification and phylogenetic placement of eukaryotic species. Here, we revisit the basic question of estimating genomic parameters such as genome length, coverage, and repeat structure, focusing specifically on estimating the $k$-mer repeat spectrum. We show using a mix of theoretical and empirical analysis that there are fundamental limitations to estimating the $k$-mer spectra due to ill-conditioned systems, and that has implications for other genomic parameters. We get around this problem using a novel constrained optimization approach (Spline Linear Programming), where the constraints are learned empirically. On reads simulated at 1X coverage from 66 genomes, our method, REPeat SPECTra Estimation (RESPECT), had 2.2% error in length estimation compared to 27% error previously achieved. In shotgun sequenced read samples with contaminants, RESPECT length estimates had median error 4%, in contrast to other methods that had median error 80%. Together, the results suggest that low-pass genomic sequencing can yield reliable estimates of the length and repeat content of the genome. The RESPECT software will be publicly available at https://urldefense.proofpoint.com/v2/url?u=https-3A__github.com_shahab-2Dsarmashghi_RESPECT.git&d=DwIGAw&c=-35OiAkTchMrZOngvJPOeA&r=ZozViWvD1E8PorCkfwYKYQMVKFoEcqLFm4Tg49XnPcA&m=f-xS8GMHKckknkc7Xpp8FJYw_ltUwz5frOw1a5pJ81EpdTOK8xhbYmrN4ZxniM96&s=717o8hLR1JmHFpRPSWG6xdUQTikyUjicjkipjFsKG4w&e=.

## Author summary

The cost of sequencing the genome is dropping at a much faster rate compared to assembling and finishing the genome. The use of lightly sampled genomes (genome skims)

SRA files used in the tests are provided in Table A in S1 Appendix.

**Funding:** VB and SS were supported in part by grants from the NSF (IIS-1815485) and from the NIH (1R01GM114362). MB, ER, and SM were supported in part by a grant from the NSF (IIS-1815485). The funders had no role in study design, data collection and analysis, decision to publish, or preparation of the manuscript.

**Competing interests:** I have read the journal's policy and the authors of this manuscript have the following competing interests: VB is a co-founder, consultant, SAB member and has equity interest in Boundless Bio, Inc. and Abterra, Inc.. The terms of this arrangement have been reviewed and approved by the University of California, San Diego in accordance with its conflict of interest policies.

could be transformative for genomic ecology. Analyzing genome skims, mostly based on statistics of small oligomers, remains challenging, but recent results have shown the advantage of this approach for the identification and phylogenetic placement of eukaryotic species. In this paper, we present a method, RESPECT, to estimate genomic properties such as genome length and repetitiveness from low-coverage genome skims. We trained RESPECT using assembled genomes and tested it on low-coverage simulated and real reads. Benchmarking results reveal that RESPECT has excellent accuracy in estimating the genome length compared to other methods, and can provide critical information regarding the repeat structure of the genome.

This is a *PLOS Computational Biology* Methods paper.

## Introduction

Anthropogenic pressure and other natural causes have resulted in severe disruption of global ecosystems in recent years, including loss of biodiversity [1]. In North America alone, the bird population has declined by over a quarter since 1970 [2]. Simply understanding the scope and extent of bio-diversity changes remains a challenging problem. Genomic sequence based bio-diversity sampling provides an attractive alternative to physical sampling and cataloging, as falling costs have made it possible to shotgun sequence a reference specimen sample for at most $10 per Gb (with another $60 for sample prep). *However, the analysis typically requires assembling and finishing a reference genome, which can still be prohibitively costly.* Despite the many projects aimed at high quality genome sequencing of eukaryotic species [3], it could be many decades before we have acquired high-quality data so that biodiversity measurements for each population can be acquired on an ongoing, routine basis.

While (meta)barcoding [4–6] methods can be used for species identification and biodiversity measurements, they have many drawbacks including limited phylogenetic resolution [7, 8]. Organelle assembly based methods [9–11] similarly cannot be used for populations and often require whole genome sequences but discard the nuclear reads (the vast majority of data). Therefore, there is renewed interest in the development of methods that use all nuclear DNA from *genome-skims*–low-coverage (0.5–2Gb) sequencing, providing 0.2–4× coverage [12]. The low coverage of skims makes them cost-effective, but insufficient for assembling, and calls for assembly-free methods. Such methods, based on analysis of $k$-mers are being actively developed [13], and have been used for species identification (Skmer [14]); for phylogenetic placement of a new species not in the library (APPLES [15]), and contaminant filtering (CONSULT [16]). While $k$-mer analysis works well for species identification, it cannot be applied easily for the analysis of populations (individuals from the same species) using genome-skims, a key component of genomic ecology. Specifically, it ignores the effect of repeats, and uses heuristics to estimate sequencing error and coverage, neither of which is known.

In this manuscript, we revisit the problem of estimating genomic parameters from genome-skim data: specifically, genome length $L$, sequence-coverage $c$, and repeat content. From genome-skim data, we have as input, abundance values of $k$-mers denoted by $\mathbf{o}$, where $o_h$ denotes the number of distinct $k$-mers of multiplicity $h$. A key latent variable is the *$k$-mer-repeat-spectrum* (denoted hereafter as the $k$-mer spectrum) of the genome described by $\mathbf{r}$,

where $r_h$ denotes the number of distinct $k$-mers that appear exactly $h$ times in the genome. As the value of $o_h$ depends upon $\mathbf{r}$, $c$, $L$, and also on sequencing error, we consider the inverse problem of estimating genomic parameters given $\mathbf{o}$ as input. The problem was studied in a seminal paper by Li and Waterman [17] who mostly considered the case of high coverage and no sequencing errors. Williams et al. [18] improved upon this model by ignoring $o_1$ assuming that a large proportion of unique $k$-mers can be attributed to sequencing errors. This assumption works better for high coverage because at low coverage, many informative $k$-mers are also seen only once. Hozza *et al.* [19] point this out, and focus attention on $k$-mer spectra. Their method, CovEst, models spectra using a geometric distribution of unknown parameters, uses that parameterized model to estimate both parameters and $r_1$, $r_2$, $r_3$, and improves estimates even for low coverage and high error.

A distinct but related line of research relates to estimating $\mathbf{o}$ itself by sub-sampling or streaming reads. Melsted and colleagues [20, 21] describe streaming algorithms to estimate $o_1$ as well as moments $F_k = \sum_i i^k o_i$. Interestingly, these moments can also be used to estimate genome parameters. For example, $\mathbb{E}[F_1] = \lambda L$, where $\lambda = (1 - (k - 1)/\ell)c$ denotes the *k-mer coverage*, or the average number of $k$-mers covering a position derived from reads of length $\ell$. We note that streaming is akin to low-coverage sampling and consider the case of estimating parameters over a range of $\lambda$.

## Estimating genome repetitiveness and other parameters using k-mers

While previous research has emphasized the estimation of genome length and coverage, we focus specifically on estimating the $k$-mer spectrum $\mathbf{r}$, defined below. Consider a genome of length $L$. Decompose the genome into a collection of all fixed-length (overlapping) sequences of length $k$, called $k$-mers. Let variables $r_j$ ($j \geq 1$) denote the number of $k$-mers that occur exactly $j$ times in the genome. When $k$ is large enough ($k \geq \log_4 L$), high values of $r_j$, for $j \geq 2$, can be attributed to the repetitive structures in the genome rather than chance similarities. Therefore, we define $\mathbf{r} = [r_1, r_2, \cdots]$ as the *(k-mer)-repeat-spectrum* of the genome.

While the repetitive sequences occur in a variety of arrangements in terms of their multiplicity, complexity and the size of repeating unit, the repeat spectrum provides a valuable summary of the extent of repetition in the genome as well as other parameters. For example, the genome length can be estimated as $L = k - 1 + \sum_j jr_j \simeq \sum_j jr_j$. Define the *uniqueness ratio* of a genome as $r_1/L$, or the ratio of the number of $k$-mers seen only once to the genome length (which is the total number of k-mers in the genome). We computed the uniqueness ratio for 622 eukaryotic genomes in RefSeq using $k = 31$ (S1 Fig). The ratio revealed a broad spectrum of values, ranging from 0.287 for *A. tauschii* (Tausch's goatgrass) to 0.995 for a mite species, *V. jacobsoni* (Fig 1A). Expectedly, there is some phylogenetic correlation and the variation of uniqueness ratio within a genus (intra-generic) is significantly lower than inter-generic variation of uniqueness ratios (S2 Fig). At higher taxonomic ranks, we observed that plants had a significantly lower uniqueness ratio compared to other groups (Fig 1B), consistent with a prevalence of whole genome duplication (WGD) events (see Methods). Nevertheless, the correlation is not strong enough to predict uniqueness ratios solely from taxonomy. For example, rice species *O. sativa* and *O. brachyantha* have different ratios 0.91 and 0.75, respectively.

The repeat spectrum provides other insights. In genomes composed largely of unique sequences, $r_1/L \simeq 1$ and $r_j$ values decrease rapidly for $j \geq 2$ with $\log r_1/r_5 \geq 4.5$ (Fig 1C). On the other hand, genomes with higher repetitive content have a smoother decrease of $r_j$ values (Fig 1D) with $\log \frac{r_1}{r_5} \leq 2.5$ (S3 Fig). Additionally, a genome that has duplicated very recently will have $r_1 \simeq 0$ and a very high value of $r_2$. Over time, however, $r_1$ increases due to the

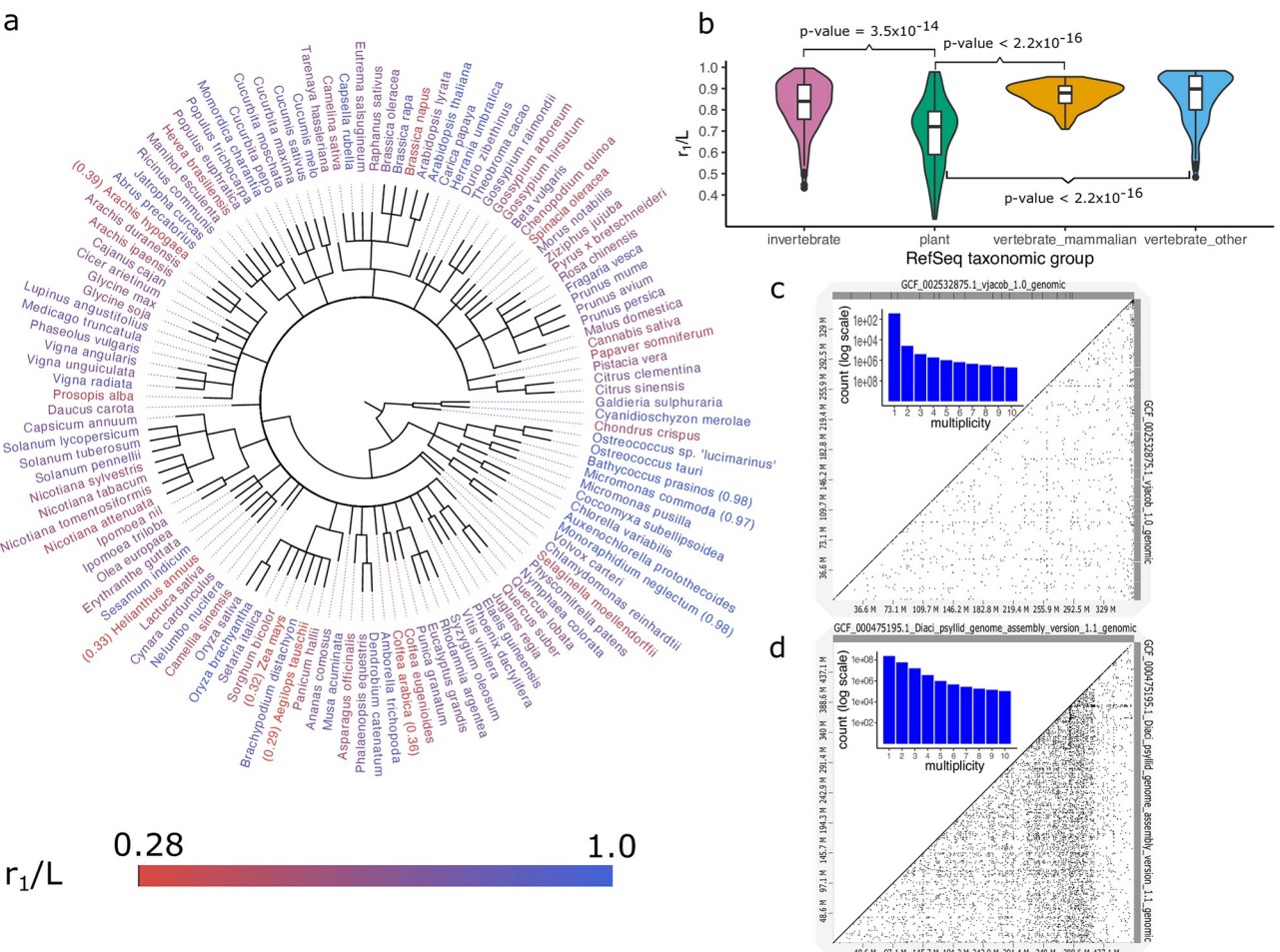

**Fig 1. Characterizing repeats at k-mer level.** A: RefSeq plant taxonomy. The species are color-coded based on the uniqueness ratio, from red (highly repetitive) to blue (non-repetitive). B: Uniqueness ratio distribution among four major taxonomic groups of eukaryotes in RefSeq. Plants (green) have significantly lower $r1/L$ compared to invertebrates (pink), mammals (yellow), and other vertebrates (blue). P-values shown on the figure, are the result of statistical tests that the uniqueness ratio is lower among plants compared to other groups. Also, to understand the extent of difference, we tested if the ratios are lower among plants by $X$% margin. The results are 5% p-value = $1.1 \times 10^{-6}$, 10% p-value = $4.3 \times 10^{-6}$, and 10% p-value = $4.2 \times 10^{-6}$ when comparing plants against invertebrates, mammals, and other vertebrates, respectively. C: Dot-plot of *V. jocobsoni* genome's (self)alignment with very few off-diagonal points, and a rapidly decaying repeat spectrum ($r_1/L$ = 0.99). D: Dot-plot of *D. citri*'s highly-repetitive genome marked by many off-diagonal elements and a smoothly decreasing repeat spectrum ($r_1/L$ = 0.51).

accumulation of mutations. Similarly, $r_j > 0$ for large values of $j$ suggest the presence of interspersed repeats.

Our method RESPECT (Repeat Spectrum identification) derives genomic length and coverage from low-coverage genome skims, while also providing insight into the repeat structure. We showed, through a mix of theoretical reasoning and empirical evidence, that the $k$-mer repeat spectra estimation problem is fundamentally difficult because of severe ill-conditioning of the system. In fact, the spectra are hard to estimate even when the coverage and sequencing error rate are known. We resolve this problem for the case of known coverage and sequencing error by imposing constraints on $r_h$ and solving a constrained optimization problem. This approach provides greatly improved estimates of **r**, which in turn lead to even better estimation of coverage, genome length and sequencing error through a stochastic iteration method. Results on genomes sampled from different parts of the tree of life and with differing repeat structures illustrate the validity of our approach.

## Results

### A simple model for estimating repeat spectra from unassembled data performs poorly

Assume that reads in the genome-skim are sequenced with a fixed mean error rate of $\epsilon$ per bp, and that the read start positions follow a Poisson distribution with a mean coverage of $\lambda$ per bp. Denote the *observed* $k$-mer data as the vector $\mathbf{o} = [o_1, o_2, \cdots]$, where $o_h$ denotes the number of $k$-mers observed exactly $h$ times in the genome-skim input. The value $o_h$ is the outcome of a random variable $O_h$ that depends upon the parameter set $\Phi = \{\lambda, \epsilon, \mathbf{r}\}$ (See Methods: 'Modeling genomic parameters'). Specifically, we assume that each $k$-mer with copy number $j$ in the genome is sampled $h$ times according to a Poisson distribution with rate dependent upon $k$, $\Phi$. Let $P_{hj}$ represent the probability of $h$ observances of a $k$-mer with copy number $j$. Then, in expectation,

$$\mathbb{E}[\mathbf{O}] = \mathbf{r}\mathbf{P^T} + \mathbf{1}_{h=1}E \tag{1}$$

where $E$ is the expected number of erroneous $k$-mers that in turn depends upon $\Phi$. $\Phi$ could be estimated using:

$$\Phi = \arg\min_{\Phi}\|\mathbf{o} - \mathbb{E}[\mathbf{O}]\| = \arg\min_{\Phi}\|\mathbf{o} - (\mathbf{r}\mathbf{P^T} + \mathbf{1}_{h=1}E)\| \tag{2}$$

In principle, an iterative procedure could be used to solve the optimization; we start with initial estimates of $\lambda$ and $\epsilon$, and use them to compute $\mathbf{P}$ and $E$. Then, we can use the least-square (LS) method to find $\mathbf{r}$ which minimizes $\|\mathbf{o} - (\mathbf{r}\mathbf{P^T} + \mathbf{1}_{h=1}E)\|$ (Eq 2) (See Methods: 'Least-squares estimate of repeat spectrum').

To study the accuracy of this model for repeat spectra estimation, we simulated genome skims at 1X coverage with no sequencing errors ($E = 0$) for all 622 genomes in RefSeq in four major taxonomic groups of eukaryotes. A subset of 66 species was selected as the test set. The test genomes were sampled such that their uniqueness-ratio ($r_1/L$) values matched the distribution of uniqueness-ratios of all 622 RefSeq genomes (S4 Fig, see Methods: 'Comparing $r1/L$ distribution over different sets'). In the following text, all parameters were trained on the 556 training genomes, and all test results shown on the 66 test genomes.

For a baseline test, we assumed that the coverage $\lambda$ was known, so that $\mathbf{r}$ could be estimated using $\|\mathbf{o} - \mathbf{r}\mathbf{P^T}\|_2$ (Eq 2). Using an LS solver (see Methods: 'Least-squares estimate of repeat spectrum'), we obtained highly accurate estimates of $r_1$ on the test data (Fig 2A; LS method). However, even in this simple case with perfect knowledge of coverage and no sequencing error, the error in estimating $r_j$ increased rapidly with increasing $j$, as the LS solution was often sparse and the estimation set $r_j = 0$ for many $j$'s, contrary to its true value in the genome.

Empirical and theoretical results showed that the poor performance could be attributed to severe ill-conditioning. We proved that the condition number of $\mathbf{P}$ grows exponentially with the number of spectra (see S1 Appendix). Therefore, small changes in $\mathbf{o}$ relative to $\mathbb{E}[\mathbf{O}]$ (Eq 1), for example due to the sampling variability or the simplifying assumptions of model, led to very large errors in estimates of $\mathbf{r}$.

### Overview of RESPECT algorithm

The negative result suggested a fundamental limitation to the use of $k$-mer based methods for estimating repeat spectra. Regularization is a proposed remedy for ill-conditioned matrices. However, most regularization methods enforce sparsity and $\mathbf{r}$ is known to be not sparse. A second challenge is that both observed counts and $k$-mer spectra are very skewed towards lower indices. Thus, a small (even 1%) relative error in $r_1$ could lead to a larger error in $r_j$ for $j > 1$.

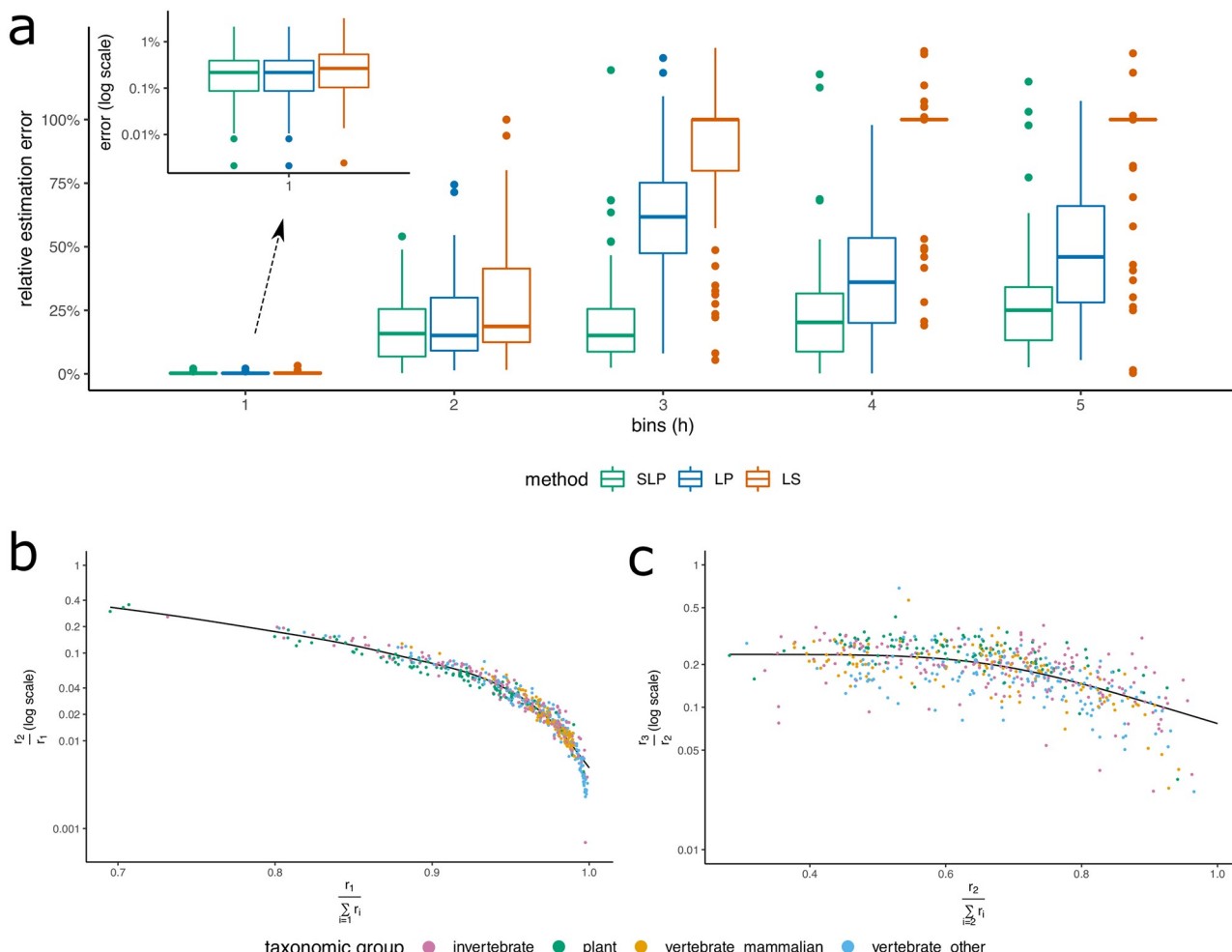

**Fig 2. Repeat spectra estimation.** A: The relative error in estimating repeat spectra using Least-Squares (LS), constrained Linear Programming (LP), and Spline Linear Programming (SLP). The genome-skims are simulated at 1X with no sequencing error. B: Correlation between true $r_2/r_1$ ratios, and our estimates of $r_1/\Sigma_{i=1} r_i$ for each genome. C: Similar correlation plot between true $r_3/r_2$ and estimated $r_2/\Sigma_{i=2} r_i$. In both B and C, true spectral ratios on Y axis are computed from the assemblies, and the estimated indices on X axis are obtained by applying the LP method to the simulated skims described in A.

To get around the ill-conditioning problem, we focused on *constraining* possible values of **r**. We observed empirically that ratio of consecutive spectral values $r_{j+1}/r_j$ was tightly constrained. Fig 2B traces $r_2/r_1$ as a function of $\frac{r_1}{\sum_{i\geq 1} r_i}$ on the training data and notes the tight correlation across all taxonomic groups. A similar, albeit less tight, constraint was observed for $r_3/r_2$ (Fig 2C) and other values as well (S5, S6 and S7 Figs).

These ideas provided the basis of a constrained linear-program for estimating **r**. As a first step, we added the constraint that $\mathcal{L}_j \leq \frac{r_j}{r_{j+1}} \leq \mathcal{U}_j$ for each $j$, where $\mathcal{L}_j$ and $\mathcal{U}_j$ are the smallest and the largest $\frac{r_j}{r_{j+1}}$ ratios over the training genomes, and solved the following LP to find **r** (see Methods: 'Linear programming for constrained optimization based estimates')

$$\mathbf{r} = \arg \min_{\mathbf{r}} \mathcal{E} = \arg \min_{\mathbf{r}} \sum_{h=2}^{n} |o_h - \sum_{j=1}^{n} P_{hj} r_j| \qquad (3)$$

This approach significantly improved the average error in estimating the spectra at multiplicity $j = 3$ and higher (Fig 2A; LP method), and resulted in small improvement at $j = 1, 2$ as well.

Using the repeat spectra from 556 training genomes, we observed a strong correlation between $r_2/r_1$ and $r_1/\sum_{i \geq 1} r_i$ (Fig 2B). Therefore, we estimated $r_2/r_1$ by using the LP estimate of $r_1/\sum_{i \geq 1} r_i$ and a spline fitted on the training data based on a generalized additive model [22, 23] (see Methods: 'Spline Linear programming'). The estimated $r_2/r_1$ value and the LP estimated $r_1$ value provided a new estimate (named SLP) of $r_2$. In a similar fashion, we computed SLP estimates of $r_{j+1}$ from LP estimate of $r_j$ and $r_j/\sum_{i \geq j} r_i$ for $j = 2, 3, 4, 5$ (Fig 2C, S5, S6 and S7 Figs, and Methods: 'Spline Linear programming'). Using the additional information learned from the training genomes captured by the fitted splines, we obtained significant reduction in the average error of repeat spectra estimation (SLP vs. LP in Fig 2A). To solve the full optimization problem in Eq 2, we used a simulated annealing procedure. Specifically, starting with initial estimates of parameters obtained under no-repeat assumption, at each iteration a new values for $\lambda$ is suggested, and SLP method is used to estimate **r**. If a candidate $\lambda$ results in a reduction in error, the algorithm accepts the move. Moreover, to avoid getting stuck at local minima, occasionally moves to states with higher error are also accepted. Lastly, the initial estimate of $\epsilon$ is corrected for the repetitiveness of genome using a regression learned over a subset of training genomes (S8 Fig). The algorithm is outlined below (also see Methods: 'RESPECT algorithm' for a detailed description).

1. Generate initial estimates of $\lambda$, $\epsilon$, and **r**.

2. Compute the initial values of **P** and error function $\mathcal{E}$.

3. For $t = 1, \cdots, N$ repeat:

    3.1. Choose $\lambda_{\text{next}}$ randomly within a neighborhood of current $\lambda$, and compute $\mathbf{P}_{\text{next}}$.

    3.2. Solve for $\mathbf{r}_{\text{next}}$ using SLP method.

    3.3. Use $\mathbf{P}_{\text{next}}$ and $\mathbf{r}_{\text{next}}$ to compute $\mathcal{E}_{\text{next}}$.

    3.4. Set $\lambda \leftarrow \lambda_{\text{next}}$, $\mathcal{E} \leftarrow \mathcal{E}_{\text{next}}$, and $\mathbf{r} \leftarrow \mathbf{r}_{\text{next}}$ with probability $\min\{1, \exp(-(\mathcal{E}_{\text{next}} - \mathcal{E})t/N)\}$.

4. Correct the initial estimate of $\epsilon$, and update $\lambda$

5. Output $c = \lambda\ell/(\ell - k + 1)$, **r**, $L = B/c$, and $\epsilon$ at the end of iterations ($B$ is the total amount of nucleotides sequenced).

## Estimating genome lengths

We applied RESPECT and CovEst to simulated genome-skims–Illumina reads sampled from the 66 test genomes skimmed at 1X coverage with 1% sequencing-error rate–and compared their relative error in the estimation of $r_1$ through $r_5$ and genome length (Fig 3), after their convergence (see S9–S14 Figs for the convergence of RESPECT's estimates). The median RESPECT error in estimating $r_1$ was less than 1.5% (average: 2.9%), while the median error of CovEst was 15% (average: 34%). The error profile extended to higher multiplicities, where, as noted earlier, CovEst used a parametric model. The tight relation between $r_1$ and $r_2$ and the large absolute differences between the two values implied that a small error in $r_1$ would translate into a large relative error for $r_2$, and we observed that for $r_2$. Similarly, the RESPECT estimates of genome length were highly accurate with median error 2.2% (average: 4.1%), in contrast to 27% (average: 40%) for CovEst (Fig 3B). RESPECT estimates were better than CovEst in 62 out of 66 species, often by considerable margins (Fig 3C). For example, in 54/66

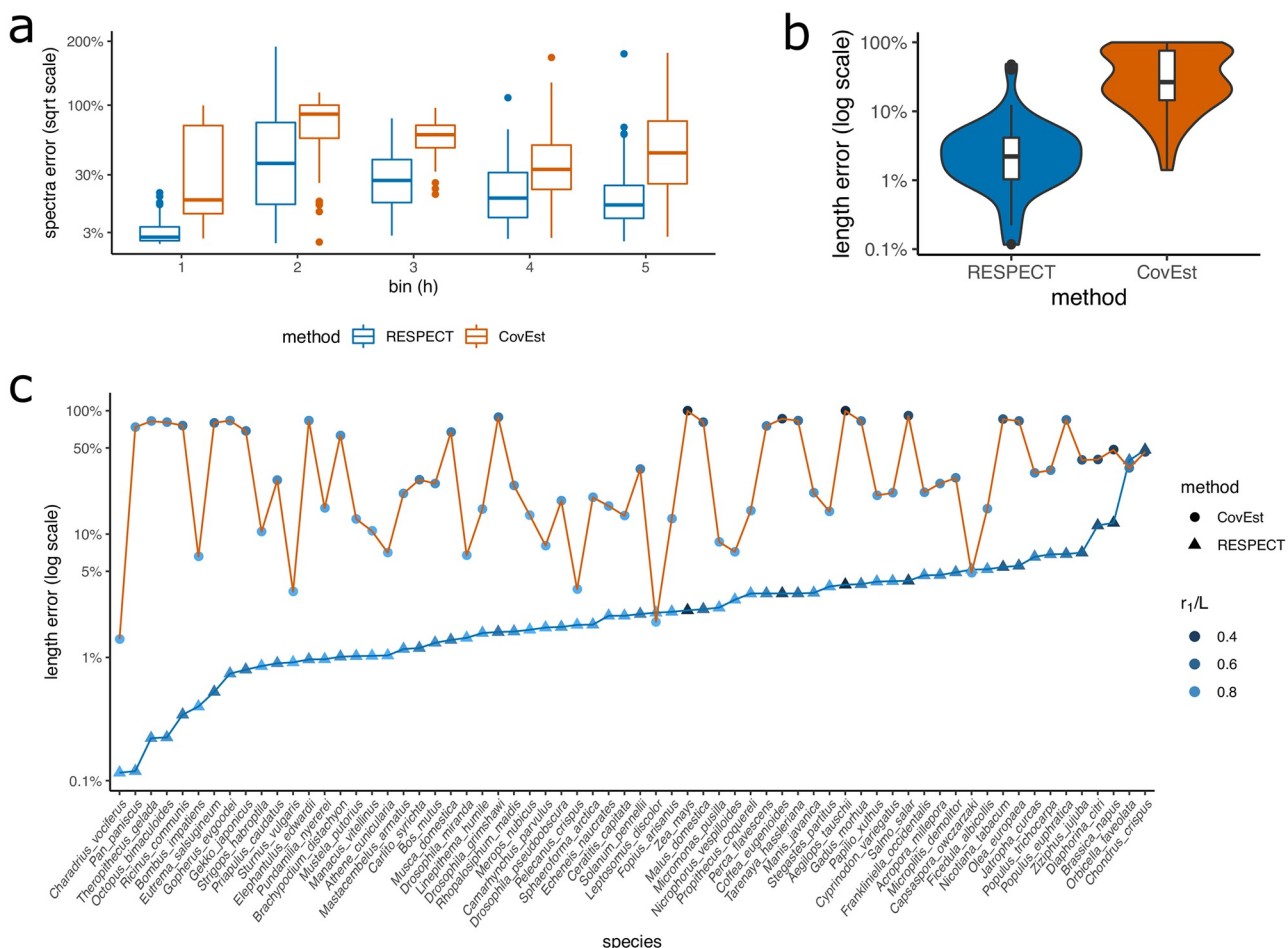

**Fig 3. Iterative estimation of genome length.** A: Comparing the error of RESPECT and CovEst in estimating the repeat spectrum. The first 5 spectra are shown. B: The distribution of error in CovEst and RESPECT. The absolute value of relative error in genome length estimation is used (in logarithmic scale). C: Per-genome error of RESPECT and CovEst in estimating the genome length of 66 species with genomes skimmed at 1X coverage.

species, RESPECT error was less than 5%, while CovEst error exceeded 50% in one third of test genomes. In fact, CovEst severely underestimates the length for these genomes (S15 Fig). For 18/66 test genomes, the CovEST estimate was less than the true length by a factor of 4 or higher (S16 Fig). RESPECT relies on models trained using available assemblies. We tested if the performance depended on the amount of training data and the taxonomic composition of the training data. RESPECT performance remained robust in these scenarios (S17(A) Fig). Moreover, its performance improved slightly (had fewer outliers) with additional training data (S17(B) Fig).

We repeated the same experiment at sequence level coverage of 0.5X, 2X, and 4X (S18 Fig). At 0.5X coverage, the median error of RESPECT was 16% (average: 18%), while CovEst had 88% median error (average: 75%) and underestimated the length by a factor of 8 or more in half of the species (S19 Fig). CovEst performance improved at higher coverage but RESPECT continued to have lower error (S20 Fig). At 4X, CovEst had median error 3.3% (average: 7.6%), while RESPECT median error was < 1% (average: 1.9%). Moreover, CovEst error exceeded 10% error in a third of species, while RESPECT had < 10% error in 64/66 species (S21 Fig).

We also compared the performance of RESPECT among different taxonomic groups. In general, plants and invertebrates had higher error rates compared to both vertebrate groups (S22 Fig), consistent with their lower uniqueness ratios (Fig 1B). In fact, we observed a statistically significant negative correlation between the estimation error and the uniqueness ratio (S23 Fig). We additionally tested RESPECT on simulated genome-skims at 1X coverage from 10 bacterial genomes, and the results did not suggest any bias against prokaryotic genomes (S24 Fig), despite the fact that we trained our model on eukaryotic genomes.

### Estimating genome length using sequenced short reads

A key difference between sequenced reads versus simulated reads is the presence of 'contaminants' or reads from non-target species. Differences may also include presence of adapter sequences, duplications of reads from the sequencing platform, lower or higher sequencing error rates due to DNA quality, and length variation of reads. Therefore, we tested RESPECT in genome-skims obtained from NCBI's Sequence Read Archive (SRA) database [24]. We downloaded high-coverage raw reads from 29 test species (from all four major taxonomic groups of eukaryotes in RefSeq) including highly repetitive plant genomes, and compared the results with the corresponding genome assemblies of the same data. After preprocessing the raw reads using BBTools [25] to remove adapter sequences and duplicate reads, we used Kraken [26] to remove contaminant reads with microbial or human origin (see Methods: 'SRA preprocessing and contamination filtering'). We note that this is an imperfect process as these tools work only when the contaminating organisms have a highly related member in the reference databases [27]. We discarded 10 samples because > 40% of reads (after removing adapters) were either duplicates of other reads, or came from external DNA sources (Table A in S1 Appendix). For the remaining 19 samples, duplicates and reads classified as contaminant were removed, and unclassified reads were sub-sampled to 1X coverage. In 16 out of 19 samples, RESPECT error was less than 11% (median: 4%), including highly repetitive genomes such as *A. tauschii* ($r_1/L = 0.29$), *Z. mays (maize)* ($r_1/L = 0.32$), *S. salar (salmon)* ($r_1/L = 0.48$), and *N. tabacum* ($r_1/L = 0.57$), where the abundance of repeats made the length estimation challenging (Fig 4, Table 1). In contrast, CovEst had less than 30% error in only 4 samples (median error 80%) (Fig 4). For the highly repetitive genomes, CovEst length estimates ranged from 1/11 to 1/7 of the assembled sequence lengths or 10 to 30 times larger error compared to RESPECT (see Table 1). In 3 samples, RESPECT had relatively high errors. For SRR085103 (domestic ferret), 99.9% of the reads did not in fact map to the available reference assembly of the domestic ferret *M. putorious*. Together with the relatively low percentage of duplication (9%) the data suggest a mislabeling of the sample species. For Coquerel's sifaka (*P. coquereli*), we observed a large gap between the total sequence length (2.8 Gbp) and the total ungapped length (2.1 Gbp) of the assembly, suggesting some challenges with the assembly. Cape elephant shrew (*E. edwardii*) was the last sample where RESPECT length estimate of 4.5Gbp exceeded the RefSeq (GCF_000299155.1) assembly length (3.8Gbp) by over 10%. Interestingly, the uniqueness ratio of the assembly was $r_1/L = 0.72$, which contrasted with the RESPECT estimated uniqueness ratio of $r_1/L = 0.65$ from the short-read data. Upon investigation, we found that a more recent assembly for *E. edwardii* (GCA_004027355.1), not yet in RefSeq, had an assembled length equal to 4.3 Gbp, with $r_1/L = 0.66$, matching the RESPECT estimates (4.5Gb, 0.65, respectively). The difference between total sequence length and ungapped length in GCA_004027355.1 was only 1 Mbp, in contrast to > 500 Mbp for GCF_000299155.1. Together, these data suggest that GCA_004027355.1 better assembles repetitive regions, and the RESPECT length estimation error was < 5%, despite using only 1X coverage.

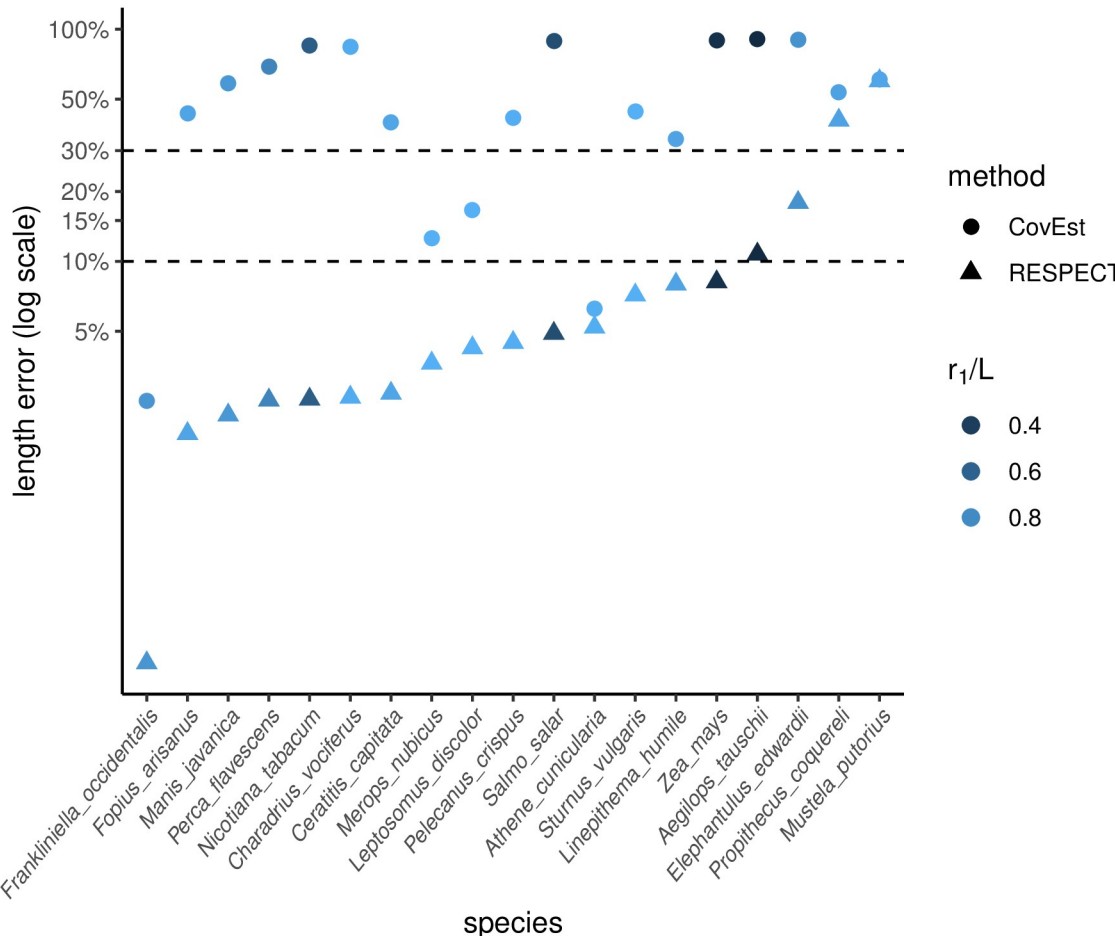

**Fig 4. Estimating genome length using SRA data.** Comparing the error of CovEst and RESPECT. High coverage SRA were preprocessed and later downsampled to 1X coverage. Both methods are applied to genome skims (after preprocessing) and the absolute values of the relative error in estimating the genome lengths are compared.

## The role of WGD versus high copy repeat elements in shaping genome repeat structure

Predicting polyploidy and recent WGD is challenging because mutation and gene loss after a WGD event can reduce the polyploidy signal. Specifically, a WGD event results in the uniqueness ratio ($r_1/L$) becoming 0. Subsequently, as mutations accumulate, $r_1/L$ ratio moves gradually towards 1 in a process that may be specific to the species, and hard to predict. Nevertheless, it should be skewed toward smaller values for recent WGD events. Independently, the presence of high copy repeats due to DNA transposons and retrotransposons can lead to very

**Table 1. Comparing RESPECT and CovEst accuracy on SRA's of highly repetitive genomes.** The numbers in parentheses are the percentage errors.

| Species | *A. tauschii* (goat grass) | *Z. mays* (maize) | *S. salar* (salmon) | *N. tabacum* (tobacco) |
|---|---|---|---|---|
| $r_1/L$ | 0.29 | 0.32 | 0.48 | 0.57 |
| Assembly length (Gbp) | 4.3 | 2.1 | 3.0 | 3.6 |
| RESPECT | 3.9 (-10.7%) | 2.0 (-8.2%) | 2.8 (-4.9%) | 3.7 (2.6%) |
| CovEst | 0.4 (-90%) | 0.2 (-90%) | 0.3 (-90%) | 0.5 (-86%) |

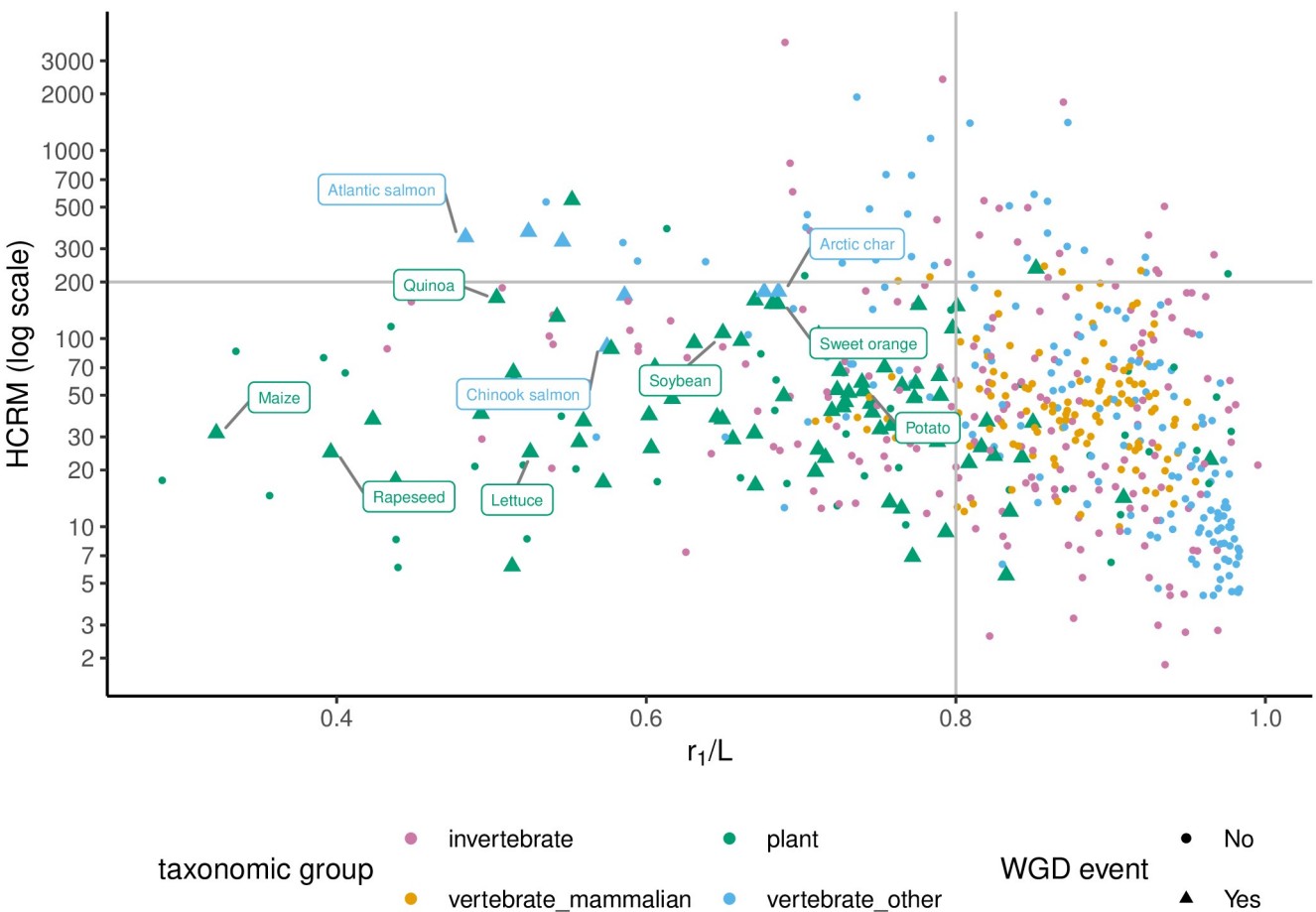

**Fig 5. High copy repeats per million versus uniqueness ratio among genomes with and without known recent WGD events.** Most of genomes with known recent WGD events had $r_1/L < 0.8$ and HCRM < 200. The y-axis is in a logarithmic scale. HCRM values are computed from genome-skims simulated at 1X coverage with no sequencing error. Some of the species with a recent WGD are labeled by their common names.

high copy numbers of a small set of oligomers. To capture the contribution of high copy repeat elements, we defined the 'High Copy Repeats per Million (HCRM)' value as the average count (per million base-pairs) of the 10 most highly repetitive k-mers. HCRM values varied across the species, ranging from 2 to 3738 among our set of 622 RefSeq genomes (S25 Fig). We observed some correlation between HCRM values of species of the same genus, especially among vertebrates (S26 Fig). However, similar to the case of uniqueness ratios, the phylogenetic signal was not pronounced enough to predict HCRM based on the taxonomy.

Analytical calculations showed that the probability of high HCRM values ≥200 in a genome with random set of k-mers was negligibly small ($P \leq 10^{-100}$) (See Methods: 'Statistical analysis of the repeat structure'), suggesting that high HCRM values could not be explained solely by WGD events, and were likely due to high copy (transposon) repeats. Fig 5 shows the ($r_1/L$, HCRM) value of 622 genome-skims, which tightly matched the true values computed from assembled genomes (S27 Fig). To analyze the $r_1/L$ and HCRM values of genomes with recent WGD, we compiled a partial list of species with known WGD events within the last 150M years based on the available literature [28–30] (See Methods: 'Selecting species with known recent WGD events' and Table B in S1 Appendix).

Species with known recent WGD events had expectedly low $r_1/L$. For example, only 14% of species with recent WGD had $r_1/L$ values $\geq 0.8$, in contrast with 64% of all species that had $r_1/L$ values higher than 0.8. Surprisingly, 93% of species with recent WGD also had low HCRM values ($\leq 200$) (Fig 5), and there was a strong association between the occurrence of recent WGD events and the ($r_1/L$, HCRM) values (p-value: $1.8 \times 10^{-23}$; See Methods: 'Statistical analysis of the repeat structure'). Our results suggest that genomes with low HCRM and $r_1/L$ are strong candidates for WGD events.

## Discussion

In this paper, we revisited the problem of estimating genomic parameters (length, sequence coverage, $k$-mer spectra) based on low coverage shotgun sequencing data. The problem has been studied previously and was considered challenging due to the need for simultaneous inference of coverage and sequencing errors along with the $k$-mer spectra. However, our results suggest that the problem remains challenging even when there is no error and the coverage is known. This is due to two factors. (a) The linear system is ill-conditioned, so that a small change in the $k$-mer counts due to random sampling can lead to large changes in the estimated $k$-mer spectra (b) Values in the $k$-mer spectra show a skewed and non-sparse distribution, where $r_1$ dominates; $r_1$ is important for length estimation, but controlling for small errors in $r_1$ leads to larger errors in the other $r_h$ values. We provide evidence of both, but future work will clarify the importance of each facet of the identification.

Proposed solutions for ill-conditioning use regularization but those methods generally enforce sparse solutions. However, the true $k$-mer distribution is not sparse. Our work resolved this issue through an empirical estimation of $k$-mer ratios based on finished genomes. This approach is viable given the many finished genomes with different repeat characteristics. Our study, with 662 genomes of which around 10% were isolated for testing, is the largest empirical study of its kind.

As expected, accurately estimated $k$-mer spectra led to better estimation of genomic parameters such as length, with RESPECT performing significantly better than the previous best method, sometimes by orders of magnitude. Our results also have lower variance than those of other methods.

As coverage increases, all methods perform well. However, at coverage 8X and higher, partial assemblies are possible and small contigs can start to be assembled. In those cases, alternative methods to estimate genome lengths may be possible, but our methods work well even for 0.5X coverage.

We had used every genome for which the assembled sequence and the raw-reads were available at the time of submission. Recently, new data has been been released, and we tested our method on 10 additional samples with very similar performance (S28 Fig).

The presence of contaminants is a significant barrier to accurate estimations, and in fact is challenging even for assembling the data. As data sampling and DNA extraction methods improve, this problem will likely be less problematic. In parallel, we are also working to improve computational approaches to removing contamination.

While most $k$-mer based statistics were developed as an initial first step prior to deep sequencing and assembly, they may have an important role to play in independent analysis of genomes. Many genomes are $\leq$ 1Gb or lower. Therefore acquiring genome-skims for a majority of organisms and even multiple individuals in a population is a feasible goal. Methods that work on these reduced representations can be transformative for studying dramatic and short-term changes in bio-ecology. We can envision technologies where a sampled individual's genome-skim can be used to quickly estimate its genome-length, repeat structure, remove

contaminating reads, identify the organism or place it confidently in the tree of life, and finally, identify the robustness of population through analysis of heterozygosity. Our paper contributes to the first step of this vision.

## Methods

### Comparing *r*1/*L* distribution over different sets

To compare two sets of values and see if the values in one set are greater than the other set, we used the Mann–Whitney *U* test. Formally, if *X* and *Y* are random samples from populations $\mathcal{X}$ and $\mathcal{Y}$, the test statistic, *U*, is given by the number of times *x* is greater than *y* for all $(x, y) \in \mathcal{X} \times \mathcal{Y}$. The Mann–Whitney *U* test is non-parametric and does not restrict the samples to be from a certain family of distributions. The test also allows the user to specify a location shift *μ* and examine the alternative hypothesis that $X - Y > \mu$. By gradually increasing *μ* and computing the p-value, we can understand the extent of difference between *X* and *Y*.

To test if two sets of numbers are drawn from the same distribution, we used the two-sample Kolmogorov–Smirnov (KS) test. The test statistic is a distance between the empirical distributions functions of the samples from the two sets. We used R 'stats' package [31] to compute the p-values for both tests.

### Modeling genomic parameters

We consider *k*-mers in a genome of length *L* and assume that $k \gg \log_4 L$ so that any *k*-mer is unlikely to appear more than once, unless it is part of a repeated sequence. Denote the (unknown) *k*-mer spectrum of a genome that contains repeats using **r**, where $r_j$ describes the number of distinct *k*-mers that appear exactly *j* times in the genome.

The genome is shotgun sequenced using reads of length $\ell$ with average sequencing depth *c*. The total number of nucleotides sequenced is given by $B = cL$. As there are $l - k + 1$ *k*-mers in each read, the *k*-mer coverage is given by

$$\lambda = (1 - (k-1)/\ell)c = \frac{(1 - (k-1)/\ell)B}{L} \ . \tag{4}$$

Let **o** denote the histogram of observed *k*-mer counts. The observed number of *k*-mers of abundance *h*, $o_h$, can be thought of as a sample allocation to random variable $O_h$, whose expected value, $m_h = \mathbb{E}[O_h]$, depends upon **r**, $\lambda$, *L*, and also on sequencing error. We assume that any base-pair is sequenced erroneously with probability $\epsilon$, and sequencing errors only result in novel *k*-mers. We further assume that the number of times a unique *k*-mer repeated *j* times is sampled follows a Poisson distribution with rate $\lambda j(1 - \epsilon)^k$. Therefore

$$\mathbf{m} = \mathbf{r}\mathbf{P}^{\mathbf{T}} + \mathbf{1}_{h=1}E \ , \tag{5}$$

where $P_{hj} = e^{-j\lambda(1-\epsilon)^k} \frac{(j\lambda(1-\epsilon)^k)^h}{h!}$ denotes the probability that a *k*-mer repeated *j* times in the genome is observed with count *h* in the genome skim, $\mathbf{1}_{h=1} = [1, 0, 0, \ldots]$, and $E = L\lambda(1 - (1 - \epsilon)^k)$ is the expected number of erroneous *k*-mers. As $\lambda$ and *L* are connected through Eq 4, we choose $\lambda$ as the independent variable and consider *L* as a function of $\lambda$. Under this model, we would like to estimate $(\mathbf{r}, \lambda, \epsilon) = \arg \min_{\mathbf{r}, \epsilon} \mathcal{E}(\mathbf{P}, \mathbf{r}, \epsilon, \mathbf{o})$, where $\mathcal{E}$ is a weighted p-norm of the difference between expected and observed counts

$$\mathcal{E}_{\mathbf{w},p}(\mathbf{P}, \mathbf{r}, \epsilon, \mathbf{o}) = \left( \sum_h w_h |m_h - o_h|^p \right)^{1/p} = \left( \sum_h w_h |(\mathbf{r}\mathbf{P}^{\mathbf{T}} + \mathbf{1}_{h=1}E)_h - o_h|^p \right)^{1/p} . \tag{6}$$

Note that the optimization is non-trivial because $\mathbf{P}$ and $E$ are functions of $(\mathbf{r}, \lambda, \epsilon)$, and must be simultaneously estimated.

## A generic iterative optimization for parameter estimation

The dimensions of $\mathbf{o}$ and $\mathbf{r}$ in Eq 5 are determined entirely by data and are not necessarily identical. However, we truncated both to a common dimension $n = 50$ for computational expediency. A generic optimization method could be described as below.

1. Generate initial estimates of $\lambda$, $\epsilon$, $L$.

2. Solve for $\mathbf{r}$ using Eq 6.

3. Use estimated $\mathbf{r}$ and grid-search to re-estimate $\lambda$, $\epsilon$.

4. Repeat step 2 onwards until the error has converged.

Step 2 is the key step in this procedure, and we devised a number of approaches to solve it.

## Least-squares estimate of repeat spectrum

Choosing $p = 2$ (Euclidean norm) and $w_h = 1$, $\forall h$ in Eq 6, the problem is turned into a Least-Squares (LS) optimization. To test an LS method for estimating $\mathbf{r}$, we considered the simplest sequencing-error-free case ($\epsilon = 0$), where coverage $\lambda$ was known. Therefore, $\mathbb{E}[\mathbf{O}] = \mathbf{m} = \mathbf{r}\mathbf{P}^{\mathbf{T}}$, where $\mathbf{P}$ is an $n \times n$ matrix with

$$P_{hj} = e^{-j\lambda} \frac{(j\lambda)^h}{h!} \ .$$

We showed (S1 Appendix) that $\mathbf{P}$ is non-singular and in the error-free case, it should be possible to use the estimate $\mathbf{r}^{(\mathbf{est})} = \mathbf{o}\mathbf{P}^{-\mathbf{T}}$. However, we observed that its effective rank was very small as $\Lambda$, $E$ each have rapidly diminishing eigenvalues. Therefore, instead of decomposing $\mathbf{P}$ and explicitly computing $\mathbf{P}^{-\mathbf{1}}$, we used the non-negative least squares (NNLS) method [32] to solve

$$\mathbf{r}^{(\mathbf{est})} = \arg\min_{\mathbf{r}} \left\| \mathbf{o} - \mathbf{r}\mathbf{P}^{\mathbf{T}} \right\|_2 \ .$$

We used nnls method from SciPy's [33] Optimize library. Unfortunately, the LS estimates were very unreliable and showed high error. In fact, we proved, for $\lambda = 1$ (see S1 Appendix), that

$$cond(\mathbf{P}) \geq \frac{2^n}{n} \ .$$

The condition number grows exponentially with $n$ suggesting a highly ill-conditioned matrix $\mathbf{P}$ where small changes in $\mathbf{o}$ from the expected values $\mathbf{m}$ would lead to large errors in estimate of $\mathbf{r}$. For these reasons, we adopted constrained optimization methods to solve for $\mathbf{r}$.

## Linear programming for constrained optimization based estimates

We used Eq 6 with $\mathbf{w} = [0, 1, 1, \ldots, 1]$ and $p = 1$ to design a Linear programming estimate of $\mathbf{r}$ as:

$$\min_{\mathbf{r}} \sum_{h=2}^{n} \left| o_h - \sum_{j=1}^{n} P_{hj} r_j \right| \ , \tag{7}$$

such that

$$\mathcal{L}_h \leq \frac{r_h}{r_{h+1}} \leq \mathcal{U}_h, \quad h = 1, 2, \cdots, n-1$$

The rationale behind setting $w_1 = 0$ was that $o_1$ contains a large number of erroneous k-mers, so we exclude it from the objective function and use the rest of the bins to estimate $\mathbf{r}$. As $\epsilon$ is not known in general, $o_1$ was used to estimate the (average) sequencing error rate, and subsequently the $k$-mer coverage $\lambda$.

The lower and upper bounds on $\frac{r_j}{r_{j+1}}$ were determined based on the distribution $R_j$ of spectral ratios in 556 training genomes, and therefore we only search for candidate solutions $\mathbf{r}$ that satisfy the constraints. Specifically, we profiled the repeat spectra of the training genomes and set $[\mathcal{L}_j, \mathcal{U}_j]$ equal to the empirical support of $R_j$ distribution, i.e., $\mathcal{L}_j$ and $\mathcal{U}_j$ are the smallest and the largest samples observed from $R_j$ over the training genomes. We use Gurobi Optimizer [34] to solve the constrained optimization problem formulated in Eq 7.

## Spline Linear programming

The final method of estimating $\mathbf{r}$ is based on the LP estimate of $\mathbf{r}$ and the splines fitted on spectral ratios $r_j/r_{j+1}$ as functions of $\frac{r_j}{\sum_{i \geq j} r_i}$. Formally, let $r_j^{\text{LP}}$ denote the LP estimate of $r_j$ by constraining the spectral ratios to be within the support of $R_j$ among the training genomes, as discussed above. For each $j \in \{1, 2, 3, 4, 5\}$, we used a generalized additive model (GAM), learned from 556 training genomes, to predict $r_j/r_{j+1}$ based on $\frac{r_j^{\text{LP}}}{\sum_{i \geq j} r_i^{\text{LP}}}$. Specifically, we model $y_j = r_j/r_{j+1}$ for different genomes as samples drawn from dependent random variable $Y_j$, which follows gamma distribution and its mean is determined by

$$g_j(\mathbb{E}[Y_j]) = s_j\left(\frac{r_j}{\sum_{i \geq j} r_i}\right), \tag{8}$$

where $g_j$ is called the link function, and $s_j$ is the smoothing spline. These functions allow us to capture nonlinear dependencies between the variables in our model. For $j = 1, 2$, we use a logarithmic link function to account for the large dynamic range of $r_j/r_{j+1}$ over the training set, and use identity link for $j = 3, 4, 5$. For each fitted GAM, we empirically set the smoothing parameter to balance the over-fitting against the goodness of fit. We used R 'mgcv' package [35] for GAM fitting.

Using the LP estimates of $r_j$'s and plugging them into Eq 8, we predict the spectral ratios. Let $y_j^{\text{SLP}}$ denote the estimate of $y_j$ using Eq 8 on previous estimates of $\mathbf{r}$. We *recursively* re-estimate $r_j$ for $j \in \{2, 3, 4, 5, 6\}$ and call them $r_j^{\text{SLP}}$:

$$r_j^{\text{SLP}} = \begin{cases} r_j^{\text{LP}} & j = 1 \text{ and } j > 6 \\ r_{j-1}^{\text{SLP}}/y_{j-1}^{\text{SLP}} & 2 \leq j \leq 6 \end{cases} \tag{9}$$

## RESPECT algorithm

For the RESPECT algorithm, we replaced the basic iterative method described above with a simulated annealing procedure outlined in Algorithm 1 to speed up the computations. To initialize the algorithm, we started with the assumptions that genome has no repeats $\mathbf{r} = [L, 0, 0, \ldots]$, and the error-free $k$-mer counts follow a Poisson distribution (Eq 5). Defining $\lambda_{\text{ef}} = \lambda(1 - \epsilon)^k$ as the

*error-free k-mer coverage*, we estimate its initial value from the ratio of observed counts

$$\lambda_{\text{ef}} = \frac{(h^* + 1)o_{h^*+1}}{o_{h^*}}, \quad \text{where} \ \ h^* = \arg\max_{h>1} o_h,$$

and set

$$\lambda = e^{-\lambda_{\text{ef}}} \frac{\lambda_{\text{ef}}^{h^*}}{h^*!} \frac{o_1}{o_{h^*}} + \lambda_{\text{ef}}(1 - e^{-\lambda_{\text{ef}}}), \ \ \epsilon = 1 - (\lambda_{\text{ef}}/\lambda)^{1/k}$$

(see S1 Appendix). The above estimate of $\epsilon$ is used throughout the algorithm, but is corrected at the end based on the estimated uniqueness ratio (described below). Using the estimate of $\lambda_{\text{ef}}$, we compute **P**, and thus the error function $\mathcal{E}$ at the start of the algorithm. For $\mathcal{E}$, we chose **w** = [0, 1, 1, ..., 1] and $p$ = 1 in Eq 6, so

$$\mathcal{E} = \sum_{h=2}^{n} \left| o_h - \sum_{j=1}^{n} P_{hj} r_j \right|$$

With the initial values of the parameters known, RESPECT runs a simulated annealing optimization until the error converges. At each iteration, a candidate $\lambda_{\text{next}}$ in $\left[\frac{1}{2}\lambda, 3\lambda\right]$ is selected uniformly at random, and $\mathbf{P}_{\text{next}}$ is computed from $\lambda_{\text{next}}(1 - \epsilon)^k$. Next, we run SLP method on $(\mathbf{o}, \mathbf{P}_{\text{next}})$ to get $\mathbf{r}_{\text{next}}$. Throughout the algorithm, we used truncated $\mathbf{o}_{1\times m}$, $\mathbf{r}_{1\times n}$, and $\mathbf{P}_{m\times n}$ where the number of spectra is fixed at $n$ = 50 (a reasonable compromise between accuracy and speed), and the number of observed counts $m = n \cdot \max(1, \lambda_{\text{ef}})$ scales proportionally with the initial estimate of error-free $k$-mer coverage. Using $(\mathbf{o}, \mathbf{P}_{\text{next}}, \mathbf{r}_{\text{next}})$, error function for the candidate state $\mathcal{E}_{\text{next}}$ is calculated. If moving to the candidate state results in a reduction in the error ($\mathcal{E}_{\text{next}} < \mathcal{E}$), the algorithm accepts the move and updates the current estimate of parameters. In addition, to help the algorithm deal with local minima and find better solutions, a simulated annealing scheme is implemented such that the algorithm probabilistically decides to move to states with higher error. Specifically, at iteration $t$, even if $\mathcal{E}_{\text{next}} > \mathcal{E}$, the algorithm accepts the move with probability $\exp(-(\mathcal{E}_{\text{next}} - \mathcal{E})t/N)$.

At the end of iterations, the initial estimate of $\epsilon$ (obtained under no-repeats assumption) is corrected based on the estimated value of $r_1/L$. The correction was learned over 120 genomes randomly selected from the training set, and applied if the estimated coverage is smaller than 1.5X. Then, $\lambda$ is re-computed based on the corrected $\epsilon$, and is used to compute the final estimates of coverage and genome length. The estimated sequencing error rate and repeat spectrum are also provided by the algorithm.

**Algorithm 1: The RESPECT method**

```
Start with λef = λ⁽⁰⁾(1 − ε)ᵏ = (h*+1)oₕ*₊₁/oₕ*, where h* = arg maxₕ>₁ oₕ;
Compute P⁽⁰⁾, ℰ⁽⁰⁾ = minᵣℰ(P⁽⁰⁾,r⁽⁰⁾,o), and r⁽⁰⁾ = arg minᵣℰ(P⁽⁰⁾,r⁽⁰⁾,o);
Find E = o₁ − ∑ⱼP₁ⱼ⁽⁰⁾rⱼ⁽⁰⁾;
Set λ⁽⁰⁾ = e⁻λef λefʰ*/h*! · o₁/oₕ* + λef(1 − e⁻λef), and compute ε from λef and λ⁽⁰⁾;
for 1 ≤ t ≤ N do
    λ⁽ᵗ⁾ ← 𝒰[½ · λ⁽ᵗ⁻¹⁾, 3 · λ⁽ᵗ⁻¹⁾];
    Use λ⁽ᵗ⁾ and ε to compute P⁽ᵗ⁾, r⁽ᵗ⁾ = arg minᵣℰ(P⁽ᵗ⁾,r⁽ᵗ⁾,o), and
    ℰ⁽ᵗ⁾ = minᵣℰ(P⁽ᵗ⁾,r⁽⁰⁾,o);
    Move to λ⁽ᵗ⁾ with probability min{1, exp(ℰ⁽ᵗ⁻¹⁾−ℰ⁽ᵗ⁾/N−t+1)};
end
Correct ε and set λ = λ⁽ᴺ⁾(1 − ε)ᵏ/(1 − εcorrected)ᵏ;
Output c = ℓ/ℓ−k+1 λ, L = B/c, εcorrected, and r⁽ᴺ⁾
```

## SRA preprocessing and contamination filtering

After downloading SRA accessions and converting them to FASTQ using SRA Toolkit [36], we used BBDuk and Dedupe from BBTools package to trim adapter sequences and remove duplicate reads. We then ran Kraken2 to remove contamination with prokaryotic or human origin. For plant and invertebrate samples, we filtered out any read that was classified to the Kraken database at 0 confidence level (very sensitive, a single matched $k$-mer is enough for the classification). For vertebrates, due to their smaller evolutionary distance to homo sapiens, we required 0.5 confidence level (more specific, half of the read's $k$-mers should match) for human classification, and 0 confidence level for everything else in the database.

## Implementation details and running time

We use 'count' and 'histo' commands from Jellyfish [37] command line tool to compute the $k$-mer histogram of input genome-skims. In each iteration of RESPECT algorithm, we solve a constrained optimization problem using the tools provided by Gurobi Python interface in 'gurobipy' package. The running time of RESPECT slowly increases with the coverage as the size of **P** (and hence the size of optimization problem at each iteration) scales with the (initial) estimate of coverage. On average, for a typical 0.5X-4X coverage of genome-skims, it takes about 2 hours for RESPECT algorithm to converge and produce the final estimate of the parameters.

## Selecting species with known recent WGD events

From the total of 83 RefSeq genomes in our database, we obtained the WGD annotation (with estimated age) for 44 plant species [29]. WGD annotations for the remaining 32 plant species in our database were based on the data provided by the 1000 plants project [30], where either the exact same species or a species from the same genus is identified to have undergone a WGD event using transcriptomic data. We also have 7 Salmonid genomes where their common ancestor is thought to have had a WGD event about 80My ago [28].

## Statistical analysis of the repeat structure

In a random genome with length $L$, there are $L - k + 1 \simeq L$ $k$-mers, and assuming the random selection of $k$-mers is uniform over the space of all $4^k$ possible $k$-mers, the probability distribution for the copy number (CN) of each $k$-mer is

$$\text{Prob}[\text{CN} = x] = \binom{L}{x}(\frac{1}{4^k})^x(1 - \frac{1}{4^k})^{L-x} .$$

For typical values of $L \sim 100 - 1000$ Mbp and $k = 31$, the conditions to use a Poisson distribution to approximate a Binomial (see e.g., Section 5.4 of [38]) are met, i.e., $L \gg 1$ and $4^{-k} \ll 1$, hence we have

$$\text{Prob}[\text{CN} = x] = e^{-L/4^k} \frac{(L/4^k)^x}{x!} .$$

If the genome subsequently undergoes $n_w$ whole genome duplication events, the genome length is multiplied by $2^{n_w}$. However, the multiplicity of each $k$-mer increases by at most $2^{n_w}$, as mutations reduce the copy number of $k$-mers. Therefore, to have an HCRM value of H, there should exist at least a $k$-mer with copy number $x \geq HL$ in the original random genome. Now,

considering that under random-genome model the selection of *any k*-mer is equally likely, we can use the union bound (see e.g., Section 1.5 of [38]) and have

$$\text{Prob}[\text{HCRM} \geq H] \quad < \sum_{\substack{\text{all} \\ k\text{-mers}}} \sum_{x=HL} e^{-L/4^k} \frac{(L/4^k)^x}{x!}$$

$$< 4^k \sum_{x=HL} e^{-L/4^k} \frac{(L/4^k)^x}{x!} \ . \tag{10}$$

We used WolframAlpha [39] to compute the bound in (10) for several values of $H$. For $H = 200$ and $L \in [100 - 1000]$ Mbp, the resulting *p*-values were less than $10^{-100}$.

To test the association between WGD events and the values of $r_1/L$ and HCRM, we used the assembled genomes of 622 RefSeq species and constructed a two by two contingency table where columns represent the species with or without an identified recent WGD, and the rows specify whether or not the genome has $r_1/L$ and HCRM values less than 0.8 and 200, respectively. We filled the table by the count of genomes that satisfied each of these four conditions, and performed a Fisher's exact test (using R 'stats' package [31]) and got the *p*-value = $1.8 \times 10^{-23}$ for the correlation between the rows and columns of the table.

## Supporting information

**S1 Appendix. Supplementary methods and data.** Detailed mathematical derivations and supplementary tables. **Table A**: SRA preprocessing results. **Table B**: List of species with recent WGD events.
(PDF)

**S1 Fig. Whole RefSeq taxonomy with $r_1/L$ annotation.** A: Plants, B: Invertebrates, C: Mammals, D: Other vertebrates.
(TIF)

**S2 Fig. Distributions of intra-generic versus inter-generic differences in $r1/L$ for pairs of RefSeq species.** A: Plants, B: Invertebrates, C: Mammals, D: Other vertebrates.
(TIF)

**S3 Fig. Correlation of $r_1/L$ with spectral ratios.** A: $r_1/r_3$ versus $r_1/L$, B: $r_1/r_5$ versus $r_1/L$.
(TIF)

**S4 Fig. Comparing the distributions of $r1/L$ among test and all RefSeq genomes.** The p-value for the hypothesis that the distributions are different using two-sided Kolmogorov–Smirnov test is 0.93. Highly-repetitive genomes are slightly over-represented in the test set.
(TIF)

**S5 Fig. Correlation between true $r_4/r_3$ and estimated $r_3/\Sigma_{i=3} r_i$.**
(TIF)

**S6 Fig. Correlation between true $r_5/r_4$ and estimated $r_4/\Sigma_{i=4} r_i$.**
(TIF)

**S7 Fig. Correlation between true $r_6/r_5$ and estimated $r_5/\Sigma_{i=5} r_i$.**
(TIF)

**S8 Fig. Correlation between the relative error in the estimated sequencing error and the uniqueness ratio.** A subset of 120 training genomes were selected as the cross-validation set, and genome-skims were simulated at 1X coverage with 1% sequencing error rate. There is a

strong correlation ($R = -0.995$) between the error in estimating $\epsilon$ and $r_1/L$ ratio. We capped the correction at 20% (red dashed line).
(TIF)

**S9 Fig. $r_1$ estimation convergence with time.**
(TIF)

**S10 Fig. $r_2$ estimation convergence with time.**
(TIF)

**S11 Fig. $r_3$ estimation convergence with time.**
(TIF)

**S12 Fig. $r_4$ estimation convergence with time.**
(TIF)

**S13 Fig. $r_5$ estimation convergence with time.**
(TIF)

**S14 Fig. Genome length convergence with time.**
(TIF)

**S15 Fig. Genome length estimation error of RESPECT and CovEst.** The coverage is 1X, and the y-axis is in square-root scale. The sign of error indicates overestimation or underestimation. The dashed lines mark the region that the absolute value of error is less than 5%.
(TIF)

**S16 Fig. Estimated to true genome length ratio.** Comparing RESPECT and CovEst over 66 test species with genomes skimmed at 1X coverage. The y-axis is plotted in log scale, and the red dashed line at $y = 1$ is the grand truth (no error). Two genomes (*A. tauschii* (0.002) and *Z. mays* (0.003)) that CovEst had extremely low estimated to true ratios were removed to improve readability.
(TIF)

**S17 Fig. Impact of training data on length estimation accuracy.** RESPECT was trained on a subset of genomes (50 of 129 mammalian genomes and 50 of 195 invertebrate genomes were removed), and the error plotted (circles) along with the error on the original training set (triangles). A: The error per genome is plotted in log scale on the y-axis. B: The distribution of error values with RESPECT trained on the subset (blue) and the entire data set (red).
(TIF)

**S18 Fig. Length estimation error on simulated data at different coverages.** The distribution of error made by RESPECT and CovEst in estimating the length of 66 test genomes skimmed at 0.5X, 1X, 2X, and 4X coverage. The y-axis is plotted in log scale.
(TIF)

**S19 Fig. Estimated to true genome length ratio.** Comparing RESPECT and CovEst over 66 test species with genomes skimmed at 0.5X coverage. The y-axis is plotted in log scale, and the red dashed line at $y = 1$ is the grand truth (no error). Four genomes (*D. grimshawi* (0.0004), *S. salar* (0.0006), *A. tauschii* (0.0012), and *Z. mays* (0.0016)) that CovEst had extremely low estimated to true ratios were removed to improve readability.
(TIF)

**S20 Fig. Estimated to true genome length ratio.** Comparing RESPECT and CovEst over 66 test species with genomes skimmed at 2X coverage. The y-axis is plotted in log scale, and the

red dashed line at $y = 1$ is the grand truth (no error).
(TIF)

**S21 Fig. Estimated to true genome length ratio.** Comparing RESPECT and CovEst over 66 test species with genomes skimmed at 4X coverage. The y-axis is plotted in log scale, and the red dashed line at $y = 1$ is the grand truth (no error). Four genomes (*D. grimshawi* (0.0004), *S. salar* (0.0006), *A. tauschii* (0.0012), and *Z. mays* (0.0016)) that CovEst had extremely low estimated to true ratios were removed to improve readability.
(TIF)

**S22 Fig. Distribution of length estimation error over four major taxonomic groups.** Significant p-values (0.05 threshold) computed using Mann-Whitney U test are added to the plot. Plants and invertebrates have higher error rates compared to vertebrates species in our test dataset.
(TIF)

**S23 Fig. Length estimation error vs. uniqueness ratio.** Negative correlation between RESPECT's error and uniqueness ratio of the genome.
(TIF)

**S24 Fig. Length estimation error for 10 bacterial genomes.** The 10 bacterial genomes were selected at random from RefSeq and genome-skims were simulated at 1X coverage. The relative error of the estimated length is plotted in log scale on the y-axis.
(TIF)

**S25 Fig. Whole RefSeq taxonomy with HCRM annotation.** Colors are based on logarithm of HCRM values for each genome. A: Plants, B: Invertebrates, C: Mammals, D: Other vertebrates.
(TIF)

**S26 Fig. Distributions of intra-generic versus inter-generic differences in HCRM for pairs of RefSeq species.** A: Plants, B: Invertebrates, C: Mammals, D: Other vertebrates.
(TIF)

**S27 Fig. High copy repeats per million versus uniqueness ratio among genomes with and without known recent WGD events.** HRCM values are computed directly from the genome assemblies.
(TIF)

**S28 Fig. Estimating genome length using SRA data.** RESPECT was test on 10 new samples (chosen at random) made available since the original submission of the manuscript. One of the samples was removed during the preprocessing due to high duplication rate. The results for the remaining 9 samples are plotted along with the original test species. Two newly added samples with high error are *Z. cesonia* and *V. riparia*. RESPECT overestimates their genome length by %28. It could be the case that the assemblies are missing some repetitive sequences (especially *V. riparia* which a has highly repetitive genome), considering that for both species there is a gap between reported total sequence length and total ungapped length.
(TIF)

## Author Contributions

**Conceptualization:** Shahab Sarmashghi, Metin Balaban, Siavash Mirarab, Vineet Bafna.

**Data curation:** Shahab Sarmashghi, Eleonora Rachtman.

**Formal analysis:** Shahab Sarmashghi, Behrouz Touri.

**Funding acquisition:** Siavash Mirarab, Vineet Bafna.

**Methodology:** Shahab Sarmashghi, Metin Balaban, Eleonora Rachtman, Behrouz Touri, Siavash Mirarab, Vineet Bafna.

**Software:** Shahab Sarmashghi.

**Supervision:** Siavash Mirarab, Vineet Bafna.

**Validation:** Shahab Sarmashghi, Behrouz Touri, Siavash Mirarab, Vineet Bafna.

**Visualization:** Shahab Sarmashghi.

**Writing – original draft:** Shahab Sarmashghi, Behrouz Touri, Siavash Mirarab, Vineet Bafna.

**Writing – review & editing:** Shahab Sarmashghi, Behrouz Touri, Siavash Mirarab, Vineet Bafna.

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
