## [Decision Letter · Decision Letter 0]

17 Jun 2021

Dear Mr. Sarmashghi,

Thank you very much for submitting your manuscript "Estimating repeat spectra and genome length from low-coverage genome skims with RESPECT" for consideration at PLOS Computational Biology. As with all papers reviewed by the journal, your manuscript was reviewed by members of the editorial board and by several independent reviewers. The reviewers appreciated the attention to an important topic. Based on the reviews, we are likely to accept this manuscript for publication, providing that you modify the manuscript according to the review recommendations.

Sincerely,

Nicola Segata

Associate Editor

PLOS Computational Biology

Sushmita Roy

Deputy Editor

PLOS Computational Biology

[LINK]

Reviewer's Responses to Questions

**Comments to the Authors:**

Reviewer #1: The authors present a new tool called RESPECT to perform estimation of genome length, coverage and repeat content from high-throughput shotgun sequencing reads. They show that their tool provides improved performance compared to other similar alternatives (namely CovEst). Although this is a tool with a highly specific utility, the authors have done a good job in describing the rationale behind their approach and providing sufficient details on how their method works. I think this will be a valuable contribution to the field. I have a few comments that I believe would make the tool even more useful and convincing.

1) In the manuscript, RESPECT is only evaluated against eukaryotic genomes. Given that the majority of sequence repositories are dominated by prokaryotic sequences, it would be very useful to understand how their tool behaves with bacterial genomes as well, at least in predicting genome length and coverage. In principle, I see no obvious reason why it should not work, aside from the fact that bacterial genomes are much less repetitive. This would also provide a more diverse sequence set to understand if the accuracy of their method is biased to particular taxonomic groups.

2) For Fig. 3 it would be useful to evaluate the length error by taxonomy. There seems to be a big range in the accuracy of their method (from 0.1% to 50%). Is their algorithm better at predicting the length of certain taxa as opposed to others?

3) I wonder if it would be possible to further improve their genome length predictions by taking into account prior knowledge on the corresponding taxa. For instance, the authors could perform a first-pass k-mer assignment of the sequence data to a database like RefSeq to determine which taxonomic group is dominated in the data. Based on this match, they would deduce that the predicted genome length would likely sit between a certain range determined by the known genomes (e.g., in RefSeq) within that taxa. This could potentially resolve some cases where the error rates are >5-10%.

4) I have some concerns regarding the test/training datasets used to evaluate their tool. I understand the authors used 66 species as the test dataset, representing the diversity of RefSeq. However, how phylogenetically balanced were the remaining 556 genomes used for training? I imagine there were clear biases for certain taxonomic groups. How did the authors account for this?

5) The number of public, sequenced datasets used for benchmarking is relatively small (29 species). Why did the authors not test a larger and more diverse set of public data? This raises the question on how representative the results shown are of what users will actually encounter in their own datasets.

6) The analysis presented in the manuscript is strictly focused on short read data (both simulated and publicly available datasets). Given the rise of long-read sequencing, it would be important to assess how their tool copes with PacBio/nanopore sequence data. At the very least, the authors should further evaluate the accuracy of RESPECT with higher error rates typical of long-read data (it seems they have only tested a 1% sequencing error rate).

7) The font size of panels C and D of Figure 1 is extremely small. Suggest increasing to improve readability.

Reviewer #2: While I didn't have any trouble installing the dependencies it would be useful to provide a conda package or docker container prior final publication. While not required for publication it would also be useful to test on sequencing platforms aside from Illumina, specifically if it also works on PacBio CCS data.

Reviewer #3: The paper presents an interesting technique for leveraging low-coverage sequencing data to estimate important characteristics of the underlying genome, namely the genome’s length and its k-mer repeat spectrum. The problem is well-motivated as the k-mer repeat spectrum is an informative statistic that can yield, for example, the genomic diversity across individuals in a population, and the ability to determine the spectrum with a low-computational cost makes it an attractive alternative to computationally expensive methods such as complete genome assembly. The presented approach is shown to improve on previous approaches to this problem.

An additional contribution of the paper is in providing both theoretical and empirical justification for why an initial optimization approach fails to accurately estimate the k-mer repeat spectrum. The paper points out that the initial approach fails because its estimate of the spectrum is sensitive to small differences in the initial sequencing data. The novel optimization approach introduced in the paper seeks to address this difficulty by imposing constraints on the estimate of the spectrum.

The paper is well-written, and the presented method works well even at < 1 coverage. As such, I recommend accepting the paper.

**Have the authors made all data and (if applicable) computational code underlying the findings in their manuscript fully available?**

Reviewer #1: Yes

Reviewer #2: Yes

Reviewer #3: Yes

PLOS authors have the option to publish the peer review history of their article (what does this mean?). If published, this will include your full peer review and any attached files.

Reviewer #1: No

Reviewer #2: No

Reviewer #3: No

Figure Files:

Data Requirements:

Reproducibility:

References:

---

## [Decision Letter · Decision Letter 1]

13 Sep 2021

Dear Mr. Sarmashghi,

We are pleased to inform you that your manuscript 'Estimating repeat spectra and genome length from low-coverage genome skims with RESPECT' has been provisionally accepted for publication in PLOS Computational Biology.

Best regards,

Nicola Segata

Associate Editor

PLOS Computational Biology

Sushmita Roy

Deputy Editor

PLOS Computational Biology

Reviewer's Responses to Questions

**Comments to the Authors:**

Reviewer #1: I thank the authors for addressing all my comments and I have no further concerns.

**Have the authors made all data and (if applicable) computational code underlying the findings in their manuscript fully available?**

Reviewer #1: Yes

PLOS authors have the option to publish the peer review history of their article (what does this mean?). If published, this will include your full peer review and any attached files.

Reviewer #1: No

---

## [Editor Report · Acceptance letter]

9 Nov 2021

PCOMPBIOL-D-21-00449R1 

Estimating repeat spectra and genome length from low-coverage genome skims with RESPECT

Dear Dr Bafna,

I am pleased to inform you that your manuscript has been formally accepted for publication in PLOS Computational Biology. Your manuscript is now with our production department and you will be notified of the publication date in due course.

With kind regards,

Olena Szabo
